# SEMI-SUPERVISED COUNTING VIA PIXEL-BY-PIXEL DENSITY DISTRIBUTION MODELLING

## ABSTRACT

This paper focuses on semi-supervised crowd counting, where only a small portion of the training data are labeled. We formulate the pixel-wise density value to regress as a probability distribution, instead of a single deterministic value. On this basis, we propose a semi-supervised crowd counting model. Firstly, we design a pixel-wise distribution matching loss to measure the differences in the pixel-wise density distributions between the prediction and the ground-truth; Secondly, we enhance the transformer decoder by using *density tokens* to specialize the forward propagations of decoders w.r.t. different density intervals; Thirdly, we design the *interleaving consistency* self-supervised learning mechanism to learn from unlabeled data efficiently. Extensive experiments on four datasets are performed to show that our method clearly outperforms the competitors by a large margin under various labeled ratio settings. *Code will be released.*

## 1 INTRODUCTION

Crowd counting (Zhang et al., 2016; Cao et al., 2018; Ma et al., 2019) is becoming increasingly important in computer vision. It has wide applications such as congestion estimation and crowd management. A lot of fully-supervised crowd counting models have been proposed, which require a large number of labeled data to train an accurate and stable model. However, considering the density of the crowd, it is laborious and time-consuming to annotate the center of each person's head in a dataset of all dense crowd images. To alleviate the requirement for large amounts of labeled data, this paper focuses on *semi-supervised counting* where only a small portion of training data are labeled (Liu et al., 2018b).

Traditional semi-supervised counting methods target density regression and then leverage self-supervised criteria (Liu et al., 2018b; 2019b) or pseudo-label generation (Sindagi et al., 2020b; Meng et al., 2021) to exploit supervision signals under unlabeled data. These methods are designed to directly generate density maps, where each pixel is associated with a definite value. However, it is still extremely difficult to learn a good model due to the uncertainty of pixel labels. Firstly, there are commonly erroneous head locations in the annotations (Wan & Chan, 2020; Bai et al., 2020); Secondly, the pseudo labels for unlabeled training data assigned by the models are pervasively noisy.

To address these challenges, we propose a new semi-supervised counting model, termed by the Pixel-by-Pixel Probability distribution modelling Network ($P^3$Net). Unlike traditional methods which generate a deterministic pixel density value, we model the targeted density value of a pixel as a probability distribution. On this premise, we contribute to semi-supervised counting in four ways.

- We propose a Pixel-wise probabilistic Distribution (PDM) loss to match the distributions of the predicted density values and the targeted ones pixel by pixel. The PDM loss, designed in line with the discrete form of the 1D Wasserstein distance, measures the cumulative gap between the predicted distribution and the ground-truth one along the density (interval) dimension. By modeling the density intervals probabilistically, our method responds well to the uncertainty in the labels. It thus surpasses traditional methods that regards the density values deterministic.
- We incorporate the transformer decoder structure with a density-token scheme to modulate the features and generate high-quality density maps. A density token encodes the semantic information of a specific density interval. In prediction, these density-specific tokens specialize the forward propagations of the decoder with respect to the corresponding density intervals.

- We create two discrete representations of the pixel-wise density probability function and shift one to be interleaved, which are modelled by a dual-branch network structure. Then we propose an inter-branch Expectation Consistency Regularization term to reconcile the expectation of the predictions made by the two branches.
- We set up new state-of-the-art performance for semi-supervised crowd counting on four challenging crowd counting datasets, i.e. UCF-QNRF (Idrees et al., 2018), JHU-Crowd++ (Sindagi et al., 2020a), ShanghaiTech A and B (Zhang et al., 2016). Our method outperforms previous state-of-the-art methods by a wide margin under all three settings of labeled ratio. Especially, on the QNRF dataset, our method achieves remarkable error reduction by over **44** in mean absolute error and **79** in mean square error under the challenging 5% label setting.

## 2  RELATED WORKS

**Fully-supervised Crowd Counting.** Early methods tackle the crowd counting problem by exhaustively detecting every individual in the image (Liu et al., 2019c) (Liu et al., 2018a). However, these methods are sensitive to occlusion and require additional annotations like bounding boxes. With the introduction of density map (Lempitsky & Zisserman, 2010), numerous CNN-based approaches are proposed to treat crowd counting as a regression problem. MCNN (Zhang et al., 2016) employs multi-column network with adaptive Gaussian kernels to extract multi-scale features. Switch-CNN (Babu Sam et al., 2017) handles the variation of crowd density by training a switch classifier to relay a patch to a particular regressor. SANet (Cao et al., 2018) proposes a local pattern consistency loss with scale aggregation modules and transposed convolutions. CSRnet (Li et al., 2018) uses dilated kernels to enlarge receptive fields and perform accurate count estimation of highly congested scenes. BL (Ma et al., 2019) introduces the loss under Bayesian assumption to calculate the expected count of pixels. Furthermore, methods based on multi-scale mechanisms (Zeng et al., 2017; Sindagi & Patel, 2019b; Ma et al., 2020), perspective estimation (Shi et al., 2019; Yan et al., 2019) and optimal transport (Wang et al., 2020a; Ma et al., 2021; Lin et al., 2021) are proposed to overcome the problem caused by large scale variations in crowd images.

Recently, to alleviate the problem of inaccurate annotations in crowd counting, a few studies begin to find solutions from quantizing the count values within each local patch into a set of intervals and learning to classify. S-DCNet proposes a classifier and a division decider to decide which sub-region should be divided and transform the open-set counting into a closed-set problem (Xiong et al., 2019). A block-wise count level classification framework is introduced to address the problems of inaccurately generated regression targets and serious sample imbalances (Liu et al., 2019a). The work (Liu et al., 2020a) proposes an adaptive mixture regression framework and leverages on local counting map to reduce the inconsistency between training targets and evaluation criteria. UEPNet (Wang et al., 2021a) uses two criteria to minimize the prediction risk and discretization errors of classification model. Our method is distinct to most existing approaches. We revisit the paradigm of density classification from the perspective of semi-supervised learning and reveal that the interleaving quantization interval has a natural consistency self-supervision mechanism.

**Semi and Weakly-Supervised Crowd Counting.** As labeling crowd images is expensive, recent studies gradually focus on semi- and weakly-supervised crowd counting. For *semi-supervised counting*, L2R (Liu et al., 2018b) introduces an auxiliary sorting task by learning containment relationships to exploit unlabeled images. A learning mechanism based on Gaussian Process-based is proposed to generate pseudo-labels for unlabeled data in (Sindagi et al., 2020b). Zhao et al. (2020) propose an active learning framework to minimize the expensive label work. IRAST (Liu et al., 2020b) leverages a set of surrogate binary segmentation tasks to exploit the underlying constraints of unlabeled data. (Meng et al., 2021) proposes a spatial uncertainty aware teacher-student framework to alleviate uncertainty from labels. (Lin et al., 2022a) proposes a density agency to construct correlations among unlabeled images. In contrast, we consider semi-supervised crowd counting as a quantitative density-interval distribution matching problem and provide a self-supervised scheme via a consistency-constrained dual-branch structure. Moreover, there are also relevant studies about *weakly supervised counting* (Yang et al., 2020; Lei et al., 2021; Sindagi & Patel, 2019a), which pay more attention to learning from coarse annotation such as image-level labels or total counts.

**Vision Transformer.** Vision Transformer (ViT) (Dosovitskiy et al., 2020) introduces the Transformer networks (Vaswani et al., 2017) to image recognition. Transformers further advances various tasks, such as object detection (Carion et al., 2020; Zhu et al., 2020; Zheng et al., 2020; Sun et al.,

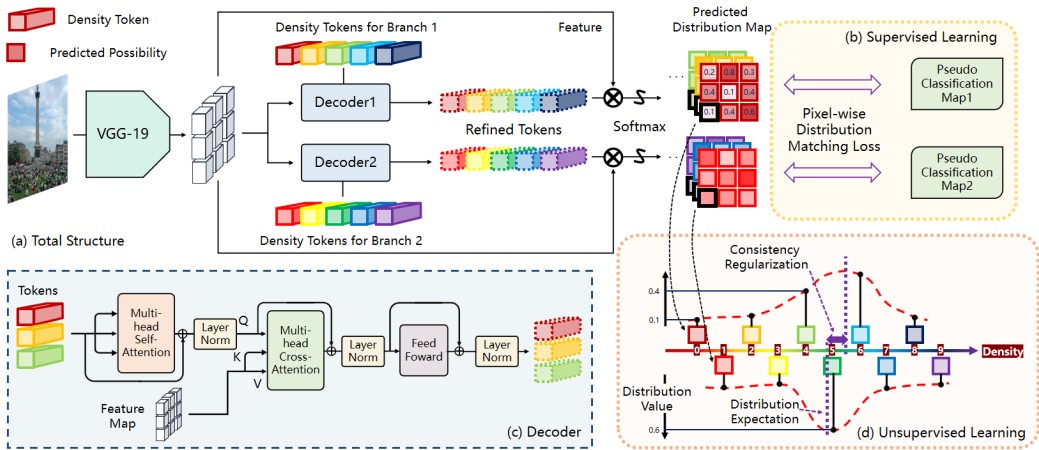

Figure 1: The structure of P³Net. (a) The interleaving dual-branch structure with density tokens to predict density category. Different token colors represent the specified different density intervals. The softmax operation targets on the category while each predicted distribution map represents the segmentation map learned by this specific token and the corresponding density category.
(b) The structure of the decoder. (c) The inter-branch Expectation Consistency Regularization for self-supervised learning. (d) The horizontal axis stands for the density values, with the attached squares for the discrete density intervals corresponding to the tokens. The vertical axis is the normalized distribution value of each category for that pixel.

2021), instance or semantic segmentation (Zheng et al., 2021; Wang et al., 2021c; Strudel et al., 2021; Cheng et al., 2021), and object tracking (Chen et al., 2021; Wang et al., 2021b; Sun et al., 2020). Lately, the works (Lin et al., 2022b; Wei et al., 2021; Liang et al., 2021) use the transformer encoder with self-attention to refine the image feature for crowd counting, whilst our method leverages the decoder with cross-attention to learn the density classification tokens.

## 3 COUNTING VIA PIXEL-BY-PIXEL PROBABILISTIC DISTRIBUTION MODELLING

In this section, we first describe the setting of semi-supervised crowd counting and then explain the rationale for adopting the probability distribution to represent the crowd density.

Formally, we have a labeled dataset $\mathcal{X}$ consisting of images with point annotated ground truth and an unlabeled dataset $\mathcal{U}$ consisting of only crowd images. In semi-supervised crowd counting, the training set includes both $\mathcal{X}$ and $\mathcal{U}$. Usually, $\mathcal{U}$ contains much more images than $\mathcal{X}$ for training a counting model, i.e., $|\mathcal{U}| \gg |\mathcal{X}|$. For crowd counting, the popular proportion settings are that the labeled dataset occupies 5%, 10% and 40% of the total training set respectively.

Previous methods have utilized self-supervised criteria (Liu et al., 2018b; 2019b) or pseudo-label generation (Sindagi et al., 2020b; Meng et al., 2021) to exploit supervision signals under unlabeled data. These methods rely on the density prediction pipeline as the most traditional supervised methods (Zhang et al., 2016; Ma et al., 2019; Lin et al., 2022b). However, when only partial labels are available for training the model, the obtained density maps are likely to be noisy. It becomes increasingly challenging to predict a deterministic, accurate density value for each single pixel or small patches. To solve this problem, we model the targeted density value of a pixel as a probability distribution, instead of a deterministic single value. The predicted density value $d$ is then given by the expectation as follows.

$$d = \int_0^{+\infty} p(x)x\,dx, \tag{1}$$

where $x$ is the probable density value ranged in $[0, +\infty)$. The conventional prediction way, which can be represented as the Dirac delta, $p(x) = \delta(x - d)$, is a special case of Eq. 1. This approach is fragile when there is uncertainty and noise. Instead, by Eq. 1, we revert to a general distribution function $p(x)$ without introducing any prior about the distribution such as Dirac.

To find a numerical form, to which deep learning models can be applied, we further discretize Eq. 1:

$$d = \sum_{j=1}^{C} P(x_j) x_j. \tag{2}$$

The set $\mathbf{v} = \{x_1, x_2, \cdots, x_C\}$ are the discrete representations of the density intervals, which are obtained by quantizing the continuous density range $[0, +\infty)$ into $C$ mutually exclusive discretized intervals $[0, b_1)$, $[b_1, b_2)$, ..., $[b_{C-1}, +\infty)$, where $b_1, ..., b_{C-1}$ are the ascending interval borders. $P(x)$ is the discrete distribution function, which can be easily implemented through a softmax function and is consistent with the convolutional neural network. As a result, in this work, we transform the regression problem into a density interval classification problem, *i.e.* from predicting an exact count to choosing a pre-defined density interval, in order to build more reliable prediction signals for semi-supervised counting.

On this basis, we propose our Pixel-by-Pixel Probability distribution modelling Network (P³Net) for semi-supervised learning. P³Net is composed of three modules to enhance the classification paradigm to semi-supervised crowd counting. First, we propose a Pixel-wise Distribution Matching (PDM) loss to meet the needs of effectively matching the distributions between the prediction and the label. After that, we introduce a transformer decoder with proposed density tokens to learn and preserve density information from different density intervals. And finally, we design a dual-branch structure and propose a corresponding self-supervision mechanism for semi-supervised learning.

### 3.1 PIXEL-WISE DISTRIBUTION MATCHING LOSS

In this section, we detail the proposed PDM loss and the corresponding supervision between predicted distribution and the ground-truth.

To punish the difference between predicted distributions and ground truth, we first generate the training label $Y \in \{0, 1\}^{N \times C}$ for the dual-branch from annotated points. We perform a 2-D Gaussian smoothing on these points, and then calculate the expected density value of each pixel. Each row in the label $\mathbf{y} \in \{0, 1\}^C$ is in the form of one hot distribution and the category where the value equals to 1 represents the specific interval that the density of this certain pixel falls into.

We match the predicted distribution to the ground-truth distribution by minimizing the divergence between them. On this basis, we adopt the Wasserstein distance to act as the measuring function. It represents the least cost of pushing one distribution $\mathbf{q}$ towards another $\tilde{\mathbf{q}}$ and is defined as:

$$W(\mathbf{q}, \tilde{\mathbf{q}}) = \min_{\pi} \int_{u,v} c(\mathbf{q}_u, \tilde{\mathbf{q}}_v) \mathrm{d}\pi(u, v). \tag{3}$$

$\pi(u, v)$ is the transport map from $\mathbf{q}_u$ to $\tilde{\mathbf{q}}_v$ while $c$ is the moving cost function. Typically, we adopt the square of Euclidean distance as $c$. We discretize the calculation of the Wasserstein distance and define the Pixel-wise Distribution Matching (PDM) loss. When both distributions are one-dimensional distribution vectors, the matching loss will have a closed-form solution (Kolouri et al., 2018). Given $\mathbf{p}$ and $\mathbf{y}$ as the prediction and ground-truth label for a certain pixel respectively, and $\mathcal{G}(\mathbf{y}, j) = \sum_{i=1}^{j} y_i$ as the cumulative distribution function, the loss can be calculated by

$$\mathcal{L}_P = \sum_{\mathbf{y}, \mathbf{p}} W(\mathbf{y}, \mathbf{p})^{\frac{1}{2}} = \sum_{\mathbf{y}, \mathbf{p}} (\sum_{j=1}^{C} (\mathcal{G}(\mathbf{y}, j) - \mathcal{G}(\mathbf{p}, j))^2)^{\frac{1}{2}}. \tag{4}$$

The PDM loss measures the cumulative gap between the predicted distribution and the ground truth along the density dimension. It penalizes the distributions that are deviated.

**The Rationale for PDM Loss.** We provide an example to illustrate the advantages of our loss function. Suppose there are four intervals and we have an instance with the label of [0,1,0,0]. Given two predicted outputs A: [0.2,0.3,0.5,0] and B: [0.2,0.3,0,0.5], clearly A gets a more compact, single-mode output which shall be considered better than B. However, the loss values of A and B are the same in terms of the Cross Entropy (0.36) and Mean Square Error (0.78), they can not be distinguished. In contrast, in terms of our PDM loss, the cumulative forms to calculate Eq. 4 for A and B are [0.2,0.5,1.0,1.0] and [0.2,0.5,0.5,1.0] respectively and the corresponding loss values are 0.29 and 0.54. As a result, the two can be well differentiated.

**Differences from DM-Count.** DM-Count (Wang et al., 2020a) is an insightful optimal transport based counting approach to match the probability distributions of occurrence over the *spatial* domain. In contrast, the proposed PDM loss matches the pixel-wise probability distributions over the *density intervals*. Hence, the domains where *optimal transport* performs by the two methods are distinctly different.

## 3.2    TRANSFORMER SPECIALIZATION

Next, we introduce a set of density tokens to specialize the forward propagations of the transformer decoder with respect to the corresponding density intervals. The density tokens are learnable embeddings with different density information, which are fed to interact with the input extracted feature vectors to instruct the model prediction. Each token is endowed with unique semantic information and acts as an indicator of a density interval. In other words, the *density tokens* are prototypes corresponding to different density intervals. Specifically, we set $b_1$ to a small value and treat the token assigned to the first interval $[0, b_1)$ as the background token. It is responsible for learning the features in areas without crowd in the image. We denote $T \in \mathbb{R}^{C \times Z}$ as a matrix capsuling all $C$ tokens where $Z$ is the dimension of both the features and tokens.

Then we use the transformer decoder (Vaswani et al., 2017) to break the limitation of local convolutional kernels, correlating similar density information from various regions inside an image. The decoder is composed of a stack of mutiple identical layers. In each decoder layer, the tokens are firstly processed by a multi-head self-attention module and a normalization layer. The relationships between tokens and the whole feature map are computed through cross attention:

$$\mathcal{C}(T, F) = \mathcal{S}(\frac{(TW^Q)(FW^K)^\mathsf{T}}{\sqrt{Z}})(FW^V). \tag{5}$$

$F \in \mathbb{R}^{N \times Z}$ is the matrix of the input features, where $N$ is the pixel or patch number. $\mathcal{S}$ is the softmax function, and $W^Q, W^K, W^V \in \mathbb{R}^{Z \times Z}$ are weight matrices for projections. Afterwards, we get the *refined tokens* $\tilde{T}$, after processing further by a layer normalization and a feed-forward network, as illustrated in Figure 1 (c).

Note that in Equation 5, through the inner product of the two vectors, the cross attention learns which regions in the feature map that each category token should focus on. Inspired by this idea, in the forward pass, we leverage the density-interval-specialized token through a softmax activation to modulate the input patch features for predicting the final probabilities:

$$O = \mathcal{S}(\tilde{T} \cdot F^\mathsf{T}), \tag{6}$$

where the softmax operation is performed along the category dimension. The predicted matrix $O \in \mathbb{R}^{C \times N}$ denotes the $C$ predicted distribution maps, each of which represents the region distribution of the corresponding density interval in the whole image of $N$ patches. By Eq. 6, we measure the similarity between the region features and refined density tokens, modulate the regional features and output the predicted density-interval distribution.

As the idea of proposed density tokens is a natural extension in semantics of that of the *query tokens* in the transformer, it can be optimized through the training pipeline of transformer using back propagation. Note that, only the original density tokens are restored, while the tokens refined adapted to the input regional features are not retained. As a result, the final density tokens are the hyper-parameters shared by all inputs in the reference stage.

**The Rationale for Tokens** arises from the observation that similar regions with same density intervals can be mined within an image. An example is shown in Figure 2, for a specific density interval like (a1), we can easily find similar regions (a2/a3) all over the image. The density tokens (a/b/c) play a role of grouping different regions with the same density levels. During learning, the tokens are connected to the discrete representation of density probability distribution and finally with clear semantic associations. During inference, by using the tokens traversally, we specialize each forward propagation of our decoder module with respect to a particular density interval distribution in turn.

**Differences from randomly-initialized queries.** Transformer decoder usually uses randomly initialized queries as the input in each forward pass. Here, we use DETR (Carion et al., 2020) as an example. In DETR, there is no clear distinction among the representative semantics of different

Figure 2: Similar Regions of the same density levels exists within an image. We use a density token to specify a density interval and group the regions of that level.

queries. Thus in the training stage, association methods like Hungarian algorithm are required in every iteration to match queries with objects. Instead, we explicitly associate an exclusive density interval to each query throughout the model's lifetime. Thus we can generate tailored tokens with clear semantics.

### 3.3 INTER-BRANCH EXPECTATION CONSISTENCY BASED ON DUAL-BRANCH INTERLEAVING STRUCTURE

Although modelling the predicted density value as a discrete probabilistic distribution leads to more credible and less noisy prediction generally, when the value falls near the boundary, noise in the output density values can easily lead to incorrect quantization, and further corrupt the classification results. Meanwhile, when converting the predicted interval category into density, the pre-defined discrete representation $x$ will inevitably have a quantization gap.

To alleviate these limitations, we use an interleaving dual branch structure, which consists of two parallel classification tasks with overlapping count intervals. The density which falls near the interval border of the first branch is more likely to be classified easily in another branch and meanwhile reducing the conversion gap without increasing the number of classification intervals.

The interleaving dual branch structure has been used to address the inaccurate ground truth (Wang et al., 2021a). To further accommodate it with semi-supervised counting, we make a step forward in this direction from two perspectives. Firstly, we associate the output of the network with the pixel-by-pixel probabilistic distribution and introduce a weighted, soft quantization level assignment mechanism. More specifically, during the inference stage, the work (Wang et al., 2021a) selects the category with maximum predicted value for each pixel or patch, and directly converts it to the corresponding representation value. Instead, we keep the distributions of predicted possibilities and leverage on the expectation to alleviate the conversion error. As a result, rather than simply averaging the predicted densities of two branches, we give each a certainty weight, which can be represented by the maximum classified possibility. When a branch predicts a large possibility for a certain category rather than similar values for multi categories, the branch has higher confidence about that prediction, thus we increase the proportion of it in the final prediction. Secondly, the interleaving two-branch structure provides a natural self-supervising mechanism that allows for imposing the consistency constraint between the two branches. On top of this constraint, we design an interleaving consistency regularization term which penalizes the deviation between the output expectations of the two-branches, to provide rich supervised signals in the absence of labels.

Specifically, we denote by two $C$-dimensional vector $\mathbf{p}, \mathbf{q} \in \mathbb{R}^C$ the predicted classified possibilities of the dual branches for a certain pixel or patch. The vectors satisfy that $\|\mathbf{p}\|_1 = \|\mathbf{q}\|_1 = 1$ and their elements are in the range of $[0, 1]$. Thus the final density can be expressed by

$$d = \omega\, \mathbf{p} \cdot \mathbf{v}_1^\mathsf{T} + (1 - \omega)\, \mathbf{q} \cdot \mathbf{v}_2^\mathsf{T}, \tag{7}$$

where the weight $\omega = \|\mathbf{p}\|_\infty / (\|\mathbf{p}\|_\infty + \|\mathbf{q}\|_\infty)$ and $\|\cdot\|_\infty$ is the vector maximum norm. $\mathbf{v}_1$ and $\mathbf{v}_2$ denote the represented quantized value for each branch. We extend the proposed network to fit the dual-branch structure, where two different decoders are adopted and the density tokens are split into two interleaved sets, as shown in Figure 1.

On the basis, a self-supervised learning scheme is designed to leverage the unlabeled data for refining the model, where the expectations of classified probability distribution on two branches tend to

be consistent. We based on this constraint to construct the inter-branch Expectation Consistency Regularization (ECR) term. Moreover, to prevent the regularization term from being negatively affected by the wrongly predicted probability distribution, we impose a selection mechanism to only consider the patches which are predicted with high certainty. The mechanism is based on a dynamic pixel-wise mask $\mathcal{E} \in \mathbb{R}^N$ which elements are in the range of $[0, 1]$ to select or weigh the regions for supervision. Given $O_1, O_2$ as the predicted probability matrices by the two branches, the self-supervised ECR is defined as

$$\mathcal{L}_E = \|\mathcal{E} \circ \mathcal{R}\|_2^2, \tag{8}$$

where $\mathcal{R} = \mathbf{v}_1 O_1 - \mathbf{v}_2 O_2$ is a vector reflecting the inconsistency between the density expectations by the two branches and $\circ$ is the element-wise multiplication.

Similar with Eq. 7, we regard the maximum possibility $\|\mathbf{p}\|_\infty$ in each distribution as the confidence. If the distribution is even, the confidence will be low, indicating that the model cannot predict a certain class for that patch with high certainty. In this case, we shall reduce its importance or exclude this patch in back-propagation dynamically. For efficient computation, we binarize $\mathcal{E} \in \{0, 1\}^N$. Only when the both confidences of two branches are sufficiently high, regularization on that pixel is activated. Given the confidence threshold $\xi \in [0, 1)$ and the boolean function $\tau(cond)$ which outputs 1 when the condition is true and 0 otherwise, the supervision mask is defined as:

$$\mathcal{E} = \tau(\mathbf{o}_1 > \xi) \& \tau(\mathbf{o}_2 > \xi), \tag{9}$$

where $\mathbf{o}_1$ and $\mathbf{o}_2$ are $N$-dimensional vectors takeing the maximum values of $O_1$ and $O_2$ along the interval dimension respectively. Finally, the overall training loss is the combination of density aware loss using in labeled data and consistency regularization with the parameter $\lambda$ using in unlabeled data.

$$\mathcal{L} = \mathcal{L}_P + \lambda \mathcal{L}_E. \tag{10}$$

**The Rationale for the Regularization.** A common issue in self-supervised consistency regularization is the confirmation bias (Tarvainen & Valpola, 2017), which indicates that the mistakes of the model will probably be accumulated during semi-supervised learning. We utilize the regularization term to alleviate this bias from the following aspects. First, we select the most reliable instances for self-supervision by using the mask in Eq. 8. Second, our network adopts two independent decoders and respective density tokens. As shown by (Ke et al., 2019), learning independent models helps to address the performance bottleneck caused by model coupling. Thus the proposed regularization term is plausible. We also provide a detailed study in the appendix.

## 4 EXPERIMENTS

We conduct extensive experiments on five crowd counting benchmarks to verify the effectiveness of proposed P³Net. Experiments and descriptions of NWPU-Crowd (Wang et al., 2020b) can be referred to the appendix. The datasets are described as follows:

**UCF-QNRF** (Idrees et al., 2018) The dataset contains congested crowd images, which are crawled from Flickr, Web Search, and Hajj footage. It includes 1,535 high-resolution images with 1.25 million annotated points. There are 1,201 and 334 images in the training and testing sets respectively.

**JHU-Crowd++ (Sindagi et al., 2020a)** The dataset includes 4,372 images with 1.51 million annotated points. There are 2,272 images used for training, 500 images for validation, and the rest 1,600 images used for testing. The crowd images are collected from several sources on the Internet using different keywords and typically chosen under various conditions and geographical locations.

**ShanghaiTech A (Zhang et al., 2016)** The dataset contains 482 crowd images with 244,167 annotated points. The images are randomly chosen from the Internet where the number of annotations in an image ranges from 33 to 3,139. The training set has 300 images, and the testing set has the remaining 182 images.

**ShanghaiTech B (Zhang et al., 2016)** The dataset contains 716 crowd images, which are taken in the crowded street of Shanghai. The number of annotations in an image ranges from 9 to 578. The training set has 316 images, and the testing set has the remaining 400 images.

The network strucutre and the trianing datails are summarized as follows.

| Methods | Labeled Percentage | UCF-QNRF | | JHU++ | | ShanghaiTech A | | ShanghaiTech B | |
|---|---|---|---|---|---|---|---|---|---|
| | | MAE | MSE | MAE | MSE | MAE | MSE | MAE | MSE |
| MT (Tarvainen & Valpola, 2017) | 5% | 172.4 | 284.9 | 101.5 | 363.5 | 104.7 | 156.9 | 19.3 | 33.2 |
| L2R (Liu et al., 2018b) | 5% | 160.1 | 272.3 | 101.4 | 338.8 | 103.0 | 155.4 | 20.3 | 27.6 |
| GP (Sindagi et al., 2020b) | 5% | 160.0 | 275.0 | - | - | 102.0 | 172.0 | 15.7 | 27.9 |
| P³Net (Ours) | 5% | **115.3** | **195.2** | **80.8** | **306.1** | **85.5** | **131.0** | **12.0** | **22.0** |
| MT (Tarvainen & Valpola, 2017) | 10% | 156.1 | 145.5 | 250.3 | 90.2 | 319.3 | 94.5 | 15.6 | 24.5 |
| L2R (Liu et al., 2018b) | 10% | 148.9 | 249.8 | 87.5 | 315.3 | 90.3 | 153.5 | 15.6 | 24.4 |
| AL-AC (Zhao et al., 2020) | 10% | - | - | - | - | 87.9 | 139.5 | 13.9 | 26.2 |
| IRAST (Liu et al., 2020b) | 10% | - | - | - | - | 86.9 | 148.9 | 14.7 | 22.9 |
| IRAST+SPN (Liu et al., 2020b) | 10% | - | - | - | - | 83.9 | 140.1 | - | - |
| P³Net (Ours) | 10% | **103.4** | **179.0** | **71.8** | **294.4** | **72.1** | **116.4** | **10.1** | **18.2** |
| MT (Tarvainen & Valpola, 2017) | 40% | 147.2 | 249.6 | 121.5 | 388.9 | 88.2 | 151.1 | 15.9 | 25.7 |
| L2R (Liu et al., 2018b) | 40% | 145.1 | 256.1 | 123.6 | 376.1 | 86.5 | 148.2 | 16.8 | 25.1 |
| GP (Sindagi et al., 2020b) | 40% | 136.0 | - | - | - | 89.0 | - | - | - |
| IRAST (Liu et al., 2020b) | 40% | 138.9 | - | - | - | - | - | - | - |
| SUA (Meng et al., 2021) | 40% | 130.3 | 226.3 | 80.7 | 290.8 | 68.5 | 121.9 | 14.1 | 20.6 |
| P³Net (Ours) | 40% | **90.0** | **155.4** | **58.9** | **251.9** | **63.0** | **100.9** | **7.1** | **12.0** |

Table 1: Comparisons with the state of the arts semi-supervised counting methods on four datasets. The best performance is shown in **bold**. The results of other methods under the 40% labeled setting are referred to (Meng et al., 2021) and all other results are from the original papers.

**Network Details.** VGG-19, which is pre-trained on ImageNet, is adopted as our CNN backbone to extract features. We use Adam algorithm (Kingma & Ba, 2014) to optimize the model with the learning rate $10^{-5}$. The number of decoder layers is set as 4. We set $C = 25$ and follow (Wang et al., 2021a) to calculate the reasonable density intervals. For the loss parameters, we set $\lambda = 0.01$ and $\xi = 0.5$.

**Training Details.** We adopt horizontal flipping and random scaling of [0.7, 1.3] for each training image. The random crop with a size of $512 \times 512$ is implemented, and as some images in Shang-haiTech A contain smaller resolution, the crop size for this dataset reduces to $256 \times 256$. We limit the shorter side of each image within 2048 pixels in all datasets. The experiments are held on one GPU of RTX3080.

## 4.1 RESULTS AND DISCUSSIONS

**Comparisons to the State of the Arts.** We evaluate P³Net on these datasets and compare it with state-of-the-art semi-supervised methods, as shown in Table 1. Since in the 50% labeled setting of paper (Meng et al., 2021), 10% of the labeled data will be used as the validation set, we consider it as the 40% labeled setting. P³Net outperforms other methods by a large margin. Compared to the second best methods on the most challenging setting of 5%, our method reduces the MAE by 44.7, 20.6, 16.5 and 3.7 points on QNRF, JHU++, ShanghaiTech A and B. Specifically, on QNRF dataset, P³Net achieves significant reductions which are over about 27.9% in mean absolute error and 28.3% in mean square error under three different settings of labeled ratio. The excellent results demonstrate the effectiveness of our method in semi-supervised crowd counting.

**The impact of PDM and ECR loss.** We conduct experiments to study the impact of two proposed loss functions. Specifically, $\mathcal{L}_P$ represents the PDM loss without ECR, and the combination of $\mathcal{L}_P$ and $\mathcal{L}_E$ forms the proposed P³Net. The comparison result is shown in Table 2. With the help of unlabeled data and the corresponding ECR, P³Net improves the counting accuracy of 'supervisions from only labeled data' over 7.8 and 12.2 in terms of MAE and MSE respectively. The experimental results validate that through the self-supervision of ECR from unlabeled data, the prediction capability and accuracy of the model is enhanced. The improvement is the sense of semi-supervised learning.

**The impact of Pixel-wise Distribution Matching loss.** We study the proposed PDM loss by comparing it with the Cross Entropy (CE) loss and MSE loss, and more noteworthy, the counting loss including the Bayesian loss (Ma et al., 2019) and DM loss (Wang et al., 2020a) which achieve best results in the fully-supervised domain. The experimental result is shown in Table 3, which is held on UCF-QNRF dataset with a labeled ratio of 5%. Our loss outperforms all four losses by large

| Labeled Percentage | Loss | MAE | MSE | Loss | MAE | MSE |
|---|---|---|---|---|---|---|
| 5% | $\mathcal{L}_P$ | 129.5 | 212.8 | $\mathcal{L}_P + \lambda\mathcal{L}_E$ | 115.3 | 195.2 |
| 10% | $\mathcal{L}_P$ | 117.4 | 211.8 | $\mathcal{L}_P + \lambda\mathcal{L}_E$ | 103.4 | 179.0 |
| 40% | $\mathcal{L}_P$ | 97.8 | 167.6 | $\mathcal{L}_P + \lambda\mathcal{L}_E$ | 90.0 | 155.4 |
| 100% | $\mathcal{L}_P$ | 78.5 | 135.8 | - | - | - |

Table 2: The impact of ECR loss. Experiments are conducted on UCF-QNRF. With the help of ECR to exploit supervisions from unlabeled data, we get a further improvement on counting accuracy.

margins. The result suggests that the awareness of the semantic information is helpful in matching the distribution between prediction and ground truth. Moreover and surprisingly, the CE loss and MSE loss, which are more specialized for classification originally, surpass the counting loss in this case. The reason probably lies in that when only a small number of ground-truth labels is available, regarding the single-value density as a probability distribution provides a better way for improving the robustness and accuracy of the counting model.

| | CE | MSE | PDM |
|---|---|---|---|
| MAE | 125.4 | 132.8 | **115.3** |
| MSE | 211.6 | 223.2 | **195.2** |

| | BL | DM | PDM |
|---|---|---|---|
| MAE | 136.5 | 133.4 | **115.3** |
| MSE | 234.7 | 225.3 | **195.2** |

| $\mathcal{L}_P^-$ | MAE | MSE |
|---|---|---|
| 5% | 134.5 | 240.6 |
| 100% | 85.8 | 142.7 |

| $\mathcal{L}_P$ | MAE | MSE |
|---|---|---|
| 5% | 129.5 | 212.8 |
| 100% | 78.5 | 135.8 |

Table 3: Comparisons of using different losses to get supervisions from ground-truth. Experiments are held on UCF-QNRF with 5% labeled ratio.

Table 4: The influence of probabilistic distribution modelling. $\mathcal{L}_P^-$ denotes the conventional prediction way, which selects the value of maximum predicted category and then a simple average is made between dual branch.

**The influence of probabilistic distribution modelling.** Table 4 reports the influence of modelling each pixel by probability distribution. For comparison, we denote the conventional prediction way as $\mathcal{L}_P^-$, which first selects the category with maximum predicted score for each pixel in each branch and converts it to the corresponding predefined representation value. Then we make a simple average instead of using Eq. 7 between dual branch. We study its performance on UCF-QNRF with the settings of 5% and 100% labeled ratios. Compared with the proposed model, the counting accuracy of $\mathcal{L}_P^-$ has an obvious decrease. This indicates that probabilistic distribution modelling and the use of expectation of different branches effectively improves the performance.

## 5 DISCUSSION AND CONCLUSION

We propose a dual-branch semi-supervised counting approach based on interleaved modelling of pixel-wise density probability distributions. The PDM loss matches the pixel-by-pixel density probability distribution to the ground truth. It shows good generalization capability, even when only a small amount of labeled data is available. Moreover, a set of tokens with clear semantic associations to the density intervals customizes the transformer decoder for the counting task. Furthermore, the inter-branch ECR term reconciles the expectations of two predicted distributions, which provides rich supervised signals for learning from unlabeled data. Our method compasses other methods by an average relative MAE reduction of over 22.0%, 23.5%, and 28.9% with the label ratios of 5%, 10%, and 40% respectively. As a result, a new strong state of the art for semi-supervised crowd counting is set up.

We also evaluate our approach under the fully-supervised setting. The detailed experimental results are reported in the appendix. Our approach achieves 78.5 MAE on QNRF and thus works remarkably well under the fully supervised setting. This consistent performance boost implies that optimal semi-supervised counting is built on both the ability to learn from labeled data and unlabeled data. Compared with those methods focusing more on learning from unlabeled data, P³Net reaches a better balance of learning from both labeled and unlabeled data. The limitation of our method is that ECR can alleviate, but cannot eliminate the bias in self-supervision. When the image background is too complex or the image is too crowded, it may lead to poor results. Nonetheless, this study still brings much inspiration for future studies about semi-supervised learning.

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
