# OpenReview forum: "Semi-supervised Counting via Pixel-by-pixel Density Distribution Modelling"
_ICLR.cc/2023/Conference — Submitted to ICLR 2023_

### Official Review · Reviewer_FHRP · 2022-10-22

**Confidence:** 3
**Correctness:** 2
**Technical Novelty And Significance:** 3
**Empirical Novelty And Significance:** 3
**Recommendation:** 6

**Clarity, Quality, Novelty And Reproducibility:**

Most of the work is innovations based on the previous work.Most of the innovations are reasonable.

**Strength And Weaknesses:**

Strength: The performance is dramaticly improved.

Weakness:
1) The N stands for the pixel number in line 189 while it means N regions in 200. It is contradictionary and with no further explanation. v1 and v2 in equation (7) is not explained.
2) How does the interleaving dual branch structure work in semi-supervied learning.
3) What's the relation between (a) and (b) in Figure1 . What's the difference in Branch 1 and 2 in Figure 1.
4) How are the density levels generated from the annotations?
5)cross attention lacks referred paper before equation (5).

**Summary Of The Paper:**

The paper is mostly clearly written. The transformer decoder and the pixel-wise distribution matching loss are adopted in the paper. The results are far better than the-state-of-art methods on four datasets which is quite good.The codes might be released for validation.

**Summary Of The Review:**

It's recommended as an acception if the problems mentioned above are clearly explained.

---

> ### Author Response · Authors · 2022-11-18
> **Response to Reviewer FHRP**
>
> Thank you very much for your great effort in reviewing our paper.
>
> **Q1**: The meaning of N and $\mathbf{v}$.
>
> **A1**: Thanks for the kind reminder. We will clarify the issues.
> Since the feature extraction backbone is generally  down-sampled, each feature in the feature map represents a region of the original image (a region of 8x8 in the paper). Thus we simply denote it as 'features for pixels' and 'features for regions'. To avoid any confusion, we will fix the issue and explain carefully in our paper.
>
> $\mathbf{v}_1$ and $\mathbf{v}_2$ represent the quantized value of intervals for each branch, which helps convert the interval category into density. $\mathbf{v}$={$x_1,x_2,...,x_c$}, where x is defined in Line 128. We will supplement the definition. Thanks.
>
> **Q2**: How does the interleaving dual branch structure work?
>
> **A2**: We shift the series of intervals (the set of events in probability)of one of the two density distribution representations to make the two series intervals interleaved. For example, branch 1 represents [0,2),[2,4),[4,6)... and branch 2 represents [0,1),[1,3),[3,5)... It is obvious that both the predictions of the two branches are the given density value and thus they shall be consistent. On this basis,  a self-supervised learning scheme which we named ECR is designed for providing extra supervision signals. The experiments show that this method is especially effective for semi-supervised counting.
>
> **Q3**: The relation between (a) and (b) in Figure1; the difference between Branch 1 and 2 in Figure 1.
>
> **A3**: Figure 1 (a) is the interleaved dual-branch network structure and (b) shows the loss (the PDM loss) calculated from the labeled data. In Figure 1 (a), we use different colors indicating the specified different density intervals. For example, in the upper branch (Branch 1), dark red indicates the density interval [0,2) while orange indicates [2,4). At the same time, the lower branch (Branch 2) will represent the interleaving density interval information, such as red indicates [0,1) and orange indicates [1,3). After a Hadamard product, each token will generate an attention map (predicted distribution map), which is specified in Eq.6, as shown on the right side of (a). Each value represents the predicted probability of the corresponding density interval at that position.
>
> In Figure 1 (b), for PDM loss on labeled data, we generate the corresponding one-hot classification labels for each branch. Since the density intervals represented by the two branches are interleaved, the corresponding ground-truth labels generated are also different, so there are two losses in (b).
>
> **Q4**: How are the density levels generated from the annotations?
>
> **A4**: The ground-truth for the counting task is in the form of discrete points. When counting how many people that are in a patch and generating the density category label for that patch, we first take the most popular density map generation solutions as (Zhang et al., 2016). We smooth each ground-truth point by a fixed-size 2D Gaussian kernel. Then we calculate the density in each patch and assign them into corresponding intervals, which are pre-defined by discretizing the whole density space.
>
> **Q5**: Cross attention lacks referred paper before equation (5).
>
> **A5**: Thanks for your reminder. We follow the structure of the decoder in the vanilla transformer (Vaswani et al., 2017), which is referred to in related works. We will follow your advice and fix the issue.

---

### Official Review · Reviewer_ccra · 2022-10-24

**Confidence:** 5
**Correctness:** 3
**Technical Novelty And Significance:** 3
**Empirical Novelty And Significance:** 3
**Recommendation:** 6

**Clarity, Quality, Novelty And Reproducibility:**

The overall expression of the manuscript is clear and of high quality, and with beautiful figures, but some mathematical expressions are not described clearly enough. The idea of introducing the idea of discrete density classification into semi-supervised population counting is also relatively novel.

**Strength And Weaknesses:**

Strengths:
(1) Transforming the density map regression task into a density-level classification task in semi-supervised crowd counting is reasonably efficient and alleviates the noise problem in the traditional semi-supervised learning of directly predicting density maps.
(2) The proposed method achieves good results on several datasets in semi-supervised crowd counting.


Weakness:
(1)In line 127 of page 4, [b1, b2] duplication occurred, which should be a clerical error.
(2)In line 215 of page 5, it is mentioned that the initialization method of density tokens in this manuscript is compared with randomly initialized queries. Ablation study about the initialization method should add to prove this assertion.
(3)In Table 1, “The results of other methods under the 40% labeled setting are referred to (Meng et al., 2021)”. However, according to (Meng et al., 2021), the quantitative results they show for the semi-supervised learning method use 50% of the labeled data. Please check to verify it.
(4)In Table 1, the methods of comparison were all proposed before 2022. If there are new articles on semi-supervised crowd counting in 2022, please cite them and compare.
(5)A visual presentation of the predicted density map compared to SOTA's semi-supervised population counting method seems to be missing in this manuscript, please add if space allows.

**Summary Of The Paper:**

In this manuscript the authors propose a semi-supervised crowd counting model, which include a pixel-wise distribution matching loss to measure the difference in pixel-wise density distribution between predicted and the ground-truth, density tokens to augment Transformer decoder, and an interleaving consistency self-supervised learning mechanism to efficiently learn from unmarked data.

**Summary Of The Review:**

Although there are still some flaws in this manuscript, the ideas are novel and the arguments are basically sufficient. I think it is a high quality manuscript.

---

> ### Author Response · Authors · 2022-11-18
> **Response to Reviewer ccra**
>
> Thank you very much for your great effort in reviewing our paper.
>
> **Q1**: Typo.
>
> **A1**: Thanks. Fixed.
>
> **Q2**: Ablation study about the initialization method.
>
> **A2**: Thanks. As per your advice, we hold the ablation experiment to study the 'randomly initialized query'. However, under the semi-supervised setting, it is not only difficult to converge, but also leads to a clear drop in accuracy. The MAE/MSE under the setting of 5% on UCF-QNRF increased to 137.2/245.7 from 115.3/195.2 respectively. We analyze this phenomenon and find that the counting task relies heavily on the density represented by each query. We will add it to the supplementary material.
>
> **Q3**: The semi-supervised setting of paper (Meng et al., 2021).
>
> **A3**: In the setting of paper (Meng et al., 2021), 10% of the labeled data (50% total) will be used as the validation set.  Thus the setting is consistent to the 40% labeled data setting in our paper. That is why we consider it as the 40% labeled setting. Thank you for the reminder, we will elaborate further in the experiment section.
>
> **Q4**: New articles on semi-supervised crowd counting in 2022.
>
> **A4**: Thanks. As per your constructive suggestion.  We have found a concurrent work [c1], which was officially published (Oct. 10) after the submission of our paper (Sept. 28). Note that our method still clearly outperforms it. We will cite it and make discussions.
>
> [c1] *Lin H, Ma Z, Hong X, et al. Semi-supervised Crowd Counting via Density Agency[C]//Proceedings of the 30th ACM International Conference on Multimedia. 2022: 1416-1426.*
>
> **Q5**: Visualizations.
>
> **A5**: Thank you for the valuable suggestion, we add the visualizations in the appendix for a clearer comparison.

---

### Official Review · Reviewer_BsbD · 2022-10-26

**Confidence:** 4
**Clarity, Quality, Novelty And Reproducibility:** No code available.
**Correctness:** 2
**Technical Novelty And Significance:** 2
**Empirical Novelty And Significance:** 2
**Recommendation:** 5

**Details Of Ethics Concerns:**

No.

**Strength And Weaknesses:**

Strength:
 Based on Table 1, it seems that this approach achieve good performance in certain benchmarks.

Weakness:
1. The idea of leveraging pixel-wise uncertainty/distribution for crowd counting is already proposed in previous work[1], which is not discussed at all.

2. The intervals mentioned in Eq.2 is highly depends on Gaussian kernel value in density generation step and the pixel-wise value of crowd density is generally very small, which makes the chosen of interval values less reliable in different settings.

3. Eq.4 is not Wasserstein distance at all, the real Wsserstein distance is an optimization process that with respect to specific constraint(transport map), thus requires iteration-based solver to get the result, while Eq.4 is a normal L2 norm. Actually, it is higly unefficient for pixel-wise Wasserstein distance as it takes extremely long time for a single forward pass.

4. How to use unlabeled images is not very clear to me.


[1]Liu, Weizhe, Nikita Durasov, and Pascal Fua. "Leveraging Self-Supervision for Cross-Domain Crowd Counting." Proceedings of the IEEE/CVF Conference on Computer Vision and Pattern Recognition. 2022.


------------------------------------------------------------------------------
Post-Rebuttal Comments:

I would like to thank the austhors provided such detailed rebuttal information, here is my comments after reading all of them:

The Wasserstein distance does have closed-form solution for 1-d formulation, but I still do not think  Eq.(4) with one-hot vector formulation is the correct formulation of 1-d Wasserstein distance as you mentioned in the reference paper. Besides, I even doubt if this formulation has any advantage over simply soft-max loss. As the code is not available, I'm not able to judge it.

I raised my recomendation considering the rebuttal did address some of my concerns.

**Summary Of The Paper:**

This paper proposes a method of semi-supervised crowd counting which only leverages a subset of labeled data and learn from the unlabeled data. This method relies on pixel-wise distribution matching and leverage optimal transport in the optimzation process.

**Summary Of The Review:**

This paper proposes a method to leverage unlabeled image by pixel-wise distribution matching, as I mentioned, I think the methodology is not sound especially the formulation of optimal transport. Therefore, I think this paper is not qualified for ICLR.

---

> ### Author Response · Authors · 2022-11-18
> **Response to Reviewer BsbD (1)**
>
> Thank you very much for your great effort in reviewing our paper.
>
> **Comment 1. The main claims of the paper are incorrect …**
>
> As per our responses to Q1-Q4 below, we disagree with the rating about ‘Correctness’ and firmly believe that our proposed approach is technically correct and our main claims are well justified.
>
> **Q1**: The idea of leveraging pixel-wise uncertainty/distribution for crowd counting … proposed in [b1].
>
> **A1**: Thanks for raising the concern. However, our method and the method mentioned are totally different.
> Firstly, optimizing the model by selecting annotations with low uncertainty (or high confidence) is widely recognized as an effective solution in semi-supervised learning and also crowd counting. For example, the paper (Liu et al., 2020b) aims to select reliable pixel-level supervision through the prior knowledge of density ranks. The work [b1] as you mentioned also selects pixel-wise pseudo-labels with low uncertainty in cross-domain learning. Though we have already cited (Liu et al., 2020b) as the work of the same kind, we will follow your advice to cite [b1] and discuss.
>
> Secondly, our paper doesn’t emphasize this aspect of selecting annotations. We focus on reformulating the crowd counting task, from a direct regression problem to a probabilistic distribution modeling problem. Note that the probability used in our method is to calculate the expectation of the density value, rather than annotation selection as [b1]. Thus the formulation in Eq. 2 makes our method completely different to [b1]. Not to mention the two loss functions PDM and ECR, which are clearly beyond the scope of previous studies [b1] and (Liu et al., 2020b).
>
> [b1] *Liu W, Durasov N, Fua P. Leveraging Self-Supervision for Cross-Domain Crowd Counting[C]//Proceedings of the IEEE/CVF Conference on Computer Vision and Pattern Recognition. 2022: 5341-5352.*
>
> **Q2**:  The intervals highly depend on Gaussian kernel value… less reliable in different settings.
>
> **A2**: Thanks for raising the concern. However, unlike the setting the intervals dependent on the Gaussian kernel (used for generating the density map) as you mentioned, our method is based on a different principle, by simply following the partition method used in (Wang et al., 2021a) to preset the interval partitions in our experiences. Note that the kernel size used in our method is 8 while in (Wang et al., 2021a) it is geometry-adaptive, which is different for each ground-truth point adaptive to the local geometry around each data point. It clearly shows that the setting is not directly linked to the Gaussian kernel size.
>
> **Q3**: Eq.4 is not the Wasserstein distance…requires iterations-based solver.
>
> **A3**: Thanks for raising the concern. We completely understand where the reviewer is coming from. This comment seems to be based on the assumption that we are calculating Wasserstein distance (WD) in Eq. 4 in a general, multi-dimensional case. However,  the distance we used in Eq. 4 is one-dimensional and it has a closed-form solution. There have been a few studies proving that there is a closed-form solution for one-dimensional WD [b2-b6], as also mentioned in Lines 157-159 in the submitted paper. It can be achieved by calculating the difference with the help of the cumulative distribution functions. In detail, since the ground truth distribution is a one-hot vector, the transport map then is fixed, i.e. all values in the predicted distribution should transport to the one-hot category.  By setting the cost of two adjacent classes c(n, n+1) as 1, the Wasserstein distance is calculated by multiplying the transport cost and the predicted value of each category, which can be written in the form of a Cumulative Distribution Function (CDF).  Then the Wasserstein distance can be obtained by summing the differences between two CDFs. Therefore, it is safe to conclude that 1d Wasserstein distance has a closed form and doesn’t necessarily use an iteration-based solver.
>
> [b2] *Frogner C, Zhang C, Mobahi H, et al. Learning with a Wasserstein loss[J]. Advances in neural information processing systems, 2015, 28.*
>
> [b3] *Ma Z, Hong X, Wei X, et al. Towards a universal model for cross-dataset crowd counting[C]//Proceedings of the IEEE/CVF International Conference on Computer Vision. 2021: 3205-3214.*
>
> [b4] *Kolouri S, Park S R, Thorpe M, et al. Optimal mass transport: Signal processing and machine-learning applications[J]. IEEE signal processing magazine, 2017, 34(4): 43-59.*
>
> [b5] *Kolouri S, Pope P E, Martin C E, et al. Sliced Wasserstein auto-encoders[C]//International Conference on Learning Representations. 2018.*
>
> [b6] *Delbracio M, Talebi H, Milanfar P. Projected distribution loss for image enhancement[J]. arXiv preprint arXiv:2012.09289, 2020.*

---

> ### Author Response · Authors · 2022-11-18
> **Response to Reviewer BsbD (2)**
>
> **Q4**: How to use unlabeled images?
>
> **A4**: We use the unlabeled training images to calculate the ECR loss in Eq. 8, which provides a self-supervised learning scheme to learn from unlabeled data. More details can be found in Section 3.3 and especially in Lines 261 to 272. The different usage of unlabeled and labeled data are shown in Figure 1 (d) and (b) respectively. Section 3.3 introduces the self-supervised learning scheme we propose for unlabeled images. We create two interleaving representations of the pixel-wise density distribution. Ideally, the expectations of classified probability distribution on two branches should be the same. Therefore we propose the inter-branch Expectation Consistency Regularization term to reconcile the expectation for unlabeled data.

---

### Official Review · Reviewer_yGdj · 2022-10-27

**Confidence:** 3
**Correctness:** 2
**Technical Novelty And Significance:** 3
**Empirical Novelty And Significance:** 4
**Recommendation:** 6

**Clarity, Quality, Novelty And Reproducibility:**

Clarity: The paper is quite clear, although some of the choices needs to be better justified, particularly the transformer architecture.

Quality: The method is well-designed and the empirical evaluation seems rigorous.

Novelty: The method is sufficiently novel.

Reproducibility: I believe there are enough details to reproduce the experiments. Code will be released.

**Details Of Ethics Concerns:**

The paper uses datasets containing images of a large number of people, mostly sourced from the internet. (Note that the paper does not introduce any new datasets, it only uses existing datasets.) It seems likely that these people did not consent to their image being used for this purpose. It's also unclear whether the datasets contain a diverse sample of people. Besides these concerns, I don't foresee any harmful impacts of the work itself, since it considers person counting rather than recognition or classification. I'm not an expert in the problem, so I'm not highly familiar with the datasets.

**Strength And Weaknesses:**

**Strengths**

1. The distribution matching loss is nice.
1. It's neat to use the uncertainty of the predicted discrete distribution to obtain the mask. I like that the non-confident examples are ignored by the ECR loss.
1. The supervised loss gives a significant improvement over the baseline methods in the fully-supervised setting (Table 5, appendix).
1. To me, it was surprising that the ECR loss, despite being simple, gave a large improvement in the semi-supervised setting (Table 2).
1. The attention map visualizations look great (appendix).

**Weaknesses**

1. The authors argue that predicting a distribution is more credible and less noisy than predicting a single value. However, the targets are always one-hot and the distribution is simply used to parametrize the prediction via its expectation, so it seems like simply an alternative parametrization and loss for predicting a single value. Another plausible explanation is that this parametrization and loss simply improve learnability and/or generalization for training a deep network.
1. The loss in eq. 4 seems more general than the loss which is actually used, since $\mathbf{y}$ is always one-hot in practice. Can the loss be simplified for one-hot targets?
1. The transformer architecture was not well justified, either by arguments or by empirical evidence. It's not clear why it's necessary. The number of categories is fixed, so couldn't we just predict logits for the categories using a conv-net? The appendix includes some variations of decoders. What about the effect of completely removing the decoders and simply comparing the learnt density tokens to the patch features?
1. The first self-attention within the decoder seems like it could be removed completely? It always takes the same set of tokens as input. Why not just learn the output tokens directly?
1. Several other ablative experiments are missing. What is the effect of using ECR loss without the mask (or varying the threshold $\xi$)? What is the effect of setting $\omega = 0.5$ instead of using the max-norm?
1. To compare the supervised loss alone to its baselines (Table 3), it seems like it would make more sense to consider the fully-supervised setting, rather than the 5% supervised setting. The fully-supervised comparison in Table 5 (appendix) does not include the CE baseline.
1. In the comparison of supervised losses (Table 3), Bayesian loss (BL) and DM loss are worse than cross entropy (CE). What could explain this?

**Minor issues**

1. It should be more clear in the main text that "semi-supervised" means that the training set is a union of fully-supervised and unsupervised sets.
1. The description of "query initialisation" seemed to refer to the fixed semantic meaning of the tokens, rather than the initial values of the query vectors, which I assume are randomly initialised and trained? This is unclear. Or if I have misunderstood, then the initialisation procedure was unclear.
1. I felt that absolute error (MAE) alone would be sufficient in the main text, and would make the tables easier to interpret. The squared error (MSE) makes the tables more cluttered and doesn't add much.
1. It wasn't clear how the 2D Gaussian smoothing was applied. Does this replace the ground-truth point annotations with a 2D Gaussian? How is the sigma chosen?
1. Why not also use the PDM loss to encourage the two discretizations to have similar outputs? (accounting for the shift)
1. The fact that the multi-head attention modules have multiple layers (4 layers) was not mentioned until the experimental details.

**Nitpicks**

* "rationale for" or "motivation of" is more appropriate than "rationality of"
* Several uses of "forward" and "forwards" as a noun. It's unclear what this means. Model evaluation?
* Some typos and grammar errors.

**Summary Of The Paper:**

This paper considers the problem of predicting density maps for counting given semi-supervised datasets.
It proposes a novel transformer-based architecture, a distribution-matching loss function for quantized density prediction, the use of two overlapping discretisations and a consistency loss for the two discretisations that is only applied for confident predictions in unsupervised images.
The method achieves strong results in both fully and partially supervised settings, achieving state-of-the-art results.
Some aspects of the design are demonstrated to be important through ablative experiments.

**Summary Of The Review:**

The paper describes an effective loss and parametrization for predicting continuous densities, as well as a simple consistency loss for unsupervised images. The transformer architecture is not demonstrated to be necessary and I wonder whether a simple conv-net would work as well using these loss functions. The ablative experiments should also be expanded, and it would be better to evaluate the supervised loss in the fully-supervised setting. I'm leaning towards accept. I may increase my rating if these issues are addressed or decrease it if they are not.

---

> ### Author Response · Authors · 2022-11-18
> **Response to Reviewer yGdj (1)**
>
> Thank you very much for your great effort in reviewing our paper.
>
> **Q1**: The rationality of predicting a distribution.
>
> **A1**: Thanks for raising the concern. Such a design is not trivial. We detail the reason as follows.
> The crowd density is of non-negative, continuous values within [0, ∞). The prediction range of the counting model can be any value from 0 to any positive value. All previous semi-supervised counting methods and most fully-supervised counting methods regard crowd counting as a regression problem and estimate a continuous density function straightforwardly. Instead, we quantize the space of [0, ∞) into a few intervals. It is conceivable that the problem of choosing a one-hot interval from a predefined set of only a few intervals is easier than predicting an exact count for an infinite value. Nevertheless, it is too coarse to just output a quantized interval. We still need additional effort in recovering a fine prediction (of density values). Thus we design such a pixel-wise distribution modeling mechanism. As we demonstrate in the experiments, our method not only outperforms other methods under the semi-supervised setting but also performs very competitively under the fully supervised setting.
>
> **Q2**: Is Eq. 4 the loss actually used? Can it be simplified?
>
> **A2**: Thanks for your constructive advice. Eq. 4 is the loss we used actually. The far right side of the equation is the L2 distance between two cumulative distribution functions namely **y** and **p**, and is already the simplified form for calculating the 1D Wasserstein distance. For example, if p is {0.1, 0.6, 0.2, 0.1} and y is {0, 1, 0, 0}. Then the cumulative distributions are {0.1, 0.7, 0.9, 1} and {0, 1, 1, 1} and the PDM loss is 0.33 by Eq. 4. For further simplification, if both **y** and **p** were one-hot, we could provide a look-up table for further acceleration. However, as p is a continuous vector of the density values, such a LUT cannot be easily built.
>
> **Q3**: The justification of the transformer architecture.
>
> **A3**: Thanks for raising the concern. The reasons for using the transformer architecture are twofold.
> The cross-attention mechanism of the transformer has been proven effective in a lot of computer vision tasks [a1-3]. Moreover,  the transformer decoder learns the unique interaction and association for each token query with full-graph features. The rationality of decoders and density tokens are explained in Section 3.2 (Line 212).
>
> As per your comments, we held an ablation study which replaces the decoder by a convolutional layer with the output dimension as the class number. Under the setting of 5% labeled data on UCF-QNRF dataset,  the MAE and MSE are 122.4 and  213.1 respectively, which are clearly inferior to the suggested one with the transformer decoder structure (MAE:115.3 and MSE: 195.2). It is safe to conclude that the cross-attention mechanism in the transformer contributes significantly to the improvement of semi-supervised counting accuracy.
>
> [a1] *Chen C F R, Fan Q, Panda R. Crossvit: Cross-attention multi-scale vision transformer for image classification[C]//Proceedings of the IEEE/CVF international conference on computer vision. 2021: 357-366.*
>
> [a2] *Carion N, Massa F, Synnaeve G, et al. End-to-end object detection with transformers[C]//European conference on computer vision. Springer, Cham, 2020: 213-229.*
>
> [a3] *Cheng B, Schwing A, Kirillov A. Per-pixel classification is not all you need for semantic segmentation[J]. Advances in Neural Information Processing Systems, 2021, 34: 17864-17875.*
>
> **Q4**: The first self-attention within the decoder.
>
> **A4**: We adopt the widely-used decoder structure in the vanilla transformer (Vaswani et al., 2017). The detailed structure is shown in Figure 1 (c). The self-attention module allows the density information to integrate across different tokens in each layer. For example, the representative features of different tokens can learn to be distinguished regularly though self-attention. Without this mechanism, each token query will be isolated and cannot interact with other queries.

---

> > ### Comment · Reviewer_yGdj · 2022-11-21
> > **Most concerns addressed, inclined to accept**
> >
> > Thanks for running the experiment using the simpler prediction head. This helps justify the transformer design. I hope you can include it in a final version.
> >
> > Thanks also for adding those ablative experiments. This updated version of Table 4 should be included in the paper, as it seems to show that only a small gain is achieved using an adaptive $\omega$.
> >
> > I'm still not convinced about the need for the initial self-attention over tokens (i.e. _before_ the feature map in the Decoder block of Figure 1). At inference, the outputs of this block will be constant, right? So we could just learn these directly?
> >
> > I think the confusion with the "initialization" stems from whether it is the keys or the values whose initialization is being discussed. Please ensure that this is clear.
> >
> > If it's standard to include both MAE and MSE in all tables, then that's ok.
> >
> > I have since discovered another relevant reference that would be good to include: Bhat et al., "AdaBins: Depth Estimation Using Adaptive Bins" (CVPR 2021). This paper considers the analogous problem of per-pixel depth estimation, similarly using the expectation of a discrete distribution to predict real values (eq. 3 in that paper).
> >
> > It seems as though the authors are correct; the Wasserstein distance has a closed form for the case of 1D distributions. However, it seems that the closed-form expression should compute the distance between inverse CDFs, whereas this paper uses the (non-inverse) CDFs (i.e. cumsums). Did I understand correctly? (Wikipedia notes a change of variables can be made _only_ for p = 1.)
> > $$
> > W_{2}^{2}(Y, P) = \int_0^1 [F_{Y}^{-1}(q) - F_{P}^{-1}(q)]^2
> > $$
> > Perhaps it is necessary to note that this is not exactly the Wasserstein distance.
> >
> > One thing that still makes me uneasy about the PDM loss is that it, when the ground-truth distribution $\mathbf{y}$ in eq. 4 is one-hot, it feels like it might be hiding a simpler measure that combines the mean and variance of the predicted distribution.
> > For example, when $Y$ is one-hot, the CDF becomes a step function $F_{Y}(q) = [q \ge c]$ and the inverse CDF becomes a constant $F_{Y}^{-1}(q) = c$.
> > The Wasserstein distance is then:
> > $$
> > W_{2}^{2}(Y, P) = \int_0^1 [F_{P}^{-1}(q) - c]^{2}
> > $$
> > However, it's not immediately obvious to me what a simpler expression would be, so I'm not counting this as a weakness of the paper.
> >
> > One last comment. The phrase "interleaving dual-branch structure" suggests that the two modules have interleaving architectures. However, the modules are effectively independent and their outputs are interleaved. I would recommend rephrasing this.
> >
> > I'm still inclined to accept this paper, provided that the ablative experiments from the rebuttal be included (Table 4 with "expectation / average" row, transformer versus conv-net).

---

> > > ### Author Response · Authors · 2022-11-21
> > > **Response to Reviewer yGdj (4)**
> > >
> > > Thank you again for your great effort in reviewing our paper.
> > >
> > > As per your advice, we will include the updated version of Table 4 and the comparison results of conv-net in the final version (since the paper cannot be revised now).
> > >
> > > **Q1**: The need of the first self-attention in decoder.
> > >
> > > **A1**: Thanks for raising this insightful concern. Exactly! You are right on the spot. The structure of the decoder (Vaswani et al., 2017) with tokens used in our paper is the same as [Carion et al., 2020; Cheng et al., 2021; a3; a4] which is composed of a stack of multiple identical layers with the same structure, so the first self-attention exists. It appears that the existing studies did not remove/replace this module and we just follow them. Nevertheless, we found that you have pointed out a very profound and constructive issue. Next we will reconsider the necessity of designing the tokens before this module in future extension work.
> > >
> > > [a4] *https://github.com/facebookresearch/detr*
> > >
> > > [a5] *https://github.com/facebookresearch/MaskFormer*
> > >
> > > **Q2**: Confusion of the term ‘initialization’.
> > >
> > > **A2**:  DETR reinitializes the token queries at each iteration.  Before each forward pass, all query vectors will be initialized to 0 and the decoder will identify the slight differences among queries by positional encoding. In contrast, we initialized once before iterations and the query vectors are learned continuously throughout the training stage. Thanks for raising this concern. To avoid confusion caused by the term ‘initialization’, we will rephrase ‘Differences from randomly-initialized queries’ to ‘Differences from reinitialized queries’.
> > >
> > > **Q3**: A good relevant reference: [Adabins, CVPR 21].
> > >
> > > **A3**: Thanks. We will cite it and discuss it in related works.
> > >
> > > **Q4**: 1D closed form of Wasserstein Distance.
> > >
> > > **A4**: Exactly! You are right. As [a6] noted, our PDM loss is strictly the Wasserstein distance if under the L1-norm. However, in the experiments conducted before, we observed that the results obtained by l2 (MAE:115.3 and MSE: 195.2) are far better than l1 (MAE:127.0 and MSE: 223.9) and we decided to use the L2 form. We analyze that this comes from the fact that l2 will have stronger supervision on the parts with larger gaps between distributions, which exactly benefits the optimization of the loss more. Thanks for your reminder. We shall follow your constructive advice and add a note that ‘the equation strictly holds under L1 norm. The one with L2 norm is a satisfactory approximation with better results in the experiments.’
> > >
> > > [a6] https://en.wikipedia.org/wiki/Wasserstein_metric#One-dimensional_distributions
> > >
> > > **Q4**: Simpler form of PDM loss.
> > >
> > > **A4**: Thanks again for the insightful and enlightening comments. We tried to follow up your first-step derivation and found that there is a $dq$ missing. The complete formula using the inverse CDFs (ICDF) with the L1 form should be
> > > $W(Y, P) = \int_0^1 |F_{P}^{-1}(q) - c| dq$.
> > >
> > > Since $F_{P}^{-1}$ is an increasing function, we further write the formula as
> > > $W(Y, P) = \int_0^\hat{q} (c - F_{P}^{-1}(q)) dq + \int_\hat{q}^1 (F_{P}^{-1}(q) - c) dq,\ {\rm where}\ \hat{q}=F_{P}(c)$.
> > >
> > > $W(Y, P) = \int_0^\hat{q} - F_{P}^{-1}(q) dq + \int_\hat{q}^1 F_{P}^{-1}(q) dq + (2\hat{q}-1)c$.
> > >
> > > However, the real difficulty lies in the conversion of ICDF and the sampling process necessary for implementation. In experiments we leveraged CDF which is convenient for calculation and followed the implementation process expressed in Eq.4. We will continue to explore in this direction to try to find a simple and general expression.
> > >
> > > **Q5**: Rephrase the term ‘Interleaving dual-branch structure’.
> > >
> > > **A5**: Thanks for your constructive advice. We see your point now. We will rephrase it by ‘Dual-branch structure with interleaving density representation’.

---

> ### Author Response · Authors · 2022-11-18
> **Response to Reviewer yGdj (2)**
>
> **Q5**: The effect of using ECR loss without the mask (or varying the threshold \xi);  the effect of setting max-norm \omega between dual branches.
>
> **A5**: Thanks for the suggestion. We held the ablation study of threshold and the results are shown below.
>
> | $\xi$ | MAE | MSE  |
> | -------- | -------- | -------- |
> | 0 | 119.8 | 199.4 |
> | 0.5 | 115.3  | 195.2 |
> | 0.75  | 117.7  | 197.4 |
>
> The counting accuracy will vary with the change of the ECR threshold, and the appropriate threshold will significantly help the counting accuracy.
>
> For $\omega$, we already study its influence in Table 4 and we further study another setting inspired by your suggestion. We calculate the expectation for each branch and make a simple average between two branches. The full experimental results are shown below.
>
> | Each Branch | Dual Branch | 5% MAE  | 5% MSE  | 100% MAE  | 100% MSE  |
> | -------- | -------- | -------- | -------- | -------- | -------- |
> | Maximum | Average | 134.5 | 240.6 | 85.8 | 142.7 |
> | Expectation | Average | 130.0 | 215.9 | 79.3 | 137.5 |
> | Expectation  | Confidence | 129.5 | 212.8  |  78.5 | 135.8 |
>
> We held experiments on UCF-QNRF under two settings: (1) only 5% labeled data without any unlabeled data; (2) 100% labeled data. This is because taking the maximum for each branch contradicts ECR, which supervised the model by expectation. For fair comparison, we only use PDM to study their difference on labeled data. Using a soft expectation value instead of a hard maximum value for each branch results in a remarkable growth in counting accuracy. Meanwhile, considering the prediction confidence when fusing between dual branches will further help improve the accuracy.
>
> **Q6**: Why does CE not exist in fully-supervised setting?
>
> **A6**: Most methods such as MCNN, SANet, BL…(Zhang et al., 2016; Cao et al., 2018; Ma et al., 2019) regard the crowd counting problem as a regression problem where CE can not work. Thus we did not list it as a baseline method for fully-supervised crowd counting. Note that only when the entire $[0, \infty]$ is discretized into a few intervals such as our method, the CE loss can be applied directly.
>
> **Q7**: More explanation of Table 3.
>
> **A7**: We study the Bayesian loss and DM loss under the proposed pixel-wise distribution modeling and transformer structure. BL and DM are specifically designed for the regression problem. And because these two losses act on the single-valued densities, in the application, we first compute the expectation of the distribution and generate the single-valued density map. Such an approach has eliminated the superiority and original motivation of the pixel-wise distribution modeling. Meanwhile CE is able to maintain this modeling and uses one hot label to give strong supervisions. On the one hand, this result comes from the fact that CE is specialized for multi-classification tasks, and on the other hand, it also justifies that with a small amount of labeled data, regarding the single-value density as a probability distribution provides a better way for improving the accuracy of the counting model.
>
> **Q8**: More introduction of semi-supervised crowd counting.
>
> **A8**: Thanks for the suggestion. In the semi-supervised setting, the training set includes two parts: a labeled dataset consisting of images with point annotated ground truth and an unlabeled dataset consisting of only crowd images. For crowd counting, the popular proportion settings are that the labeled dataset occupies 5%, 10% and 40% of the total training set respectively.  We will explain this setting in the paper.
>
> **Q9**: More explanation of random initialization.
>
> **A9**: 'Randomly initialized query' means that when each image is input to the model, the initial value of the query will be defined as 0 or random number. And the position encodings for query may be added. DETR adopts this solution in order to avoid overfitting a query to a specific object in detection. Conversely, we want to preserve the semantics of the density categories. We hold the ablation experiment to study the 'randomly initialized query'. However, under the semi-supervised setting, it is not only difficult to converge, but also leads to a clear drop in accuracy. The MAE/MSE under the setting of 5% on UCF-QNRF increased to 137.2/245.7 from 115.3/195.2 respectively. We analyze this phenomenon and find that the counting task relies heavily on the density represented by each query. We will add it to the supplementary material.
>
> **Q10**: MAE alone.
>
> **A10**: Your comments are constructive. However, the current mainstream counting papers regard the MAE and MSE as two important indicators. We have followed this reported way.

---

> ### Author Response · Authors · 2022-11-18
> **Response to Reviewer yGdj (3)**
>
> **Q11**: More explanation of 2D Gaussian smoothing.
>
> **A11**: The ground-truth for the counting task is in the form of discrete points. When counting how many people that are in a patch and generating the density category label for that patch, we first take the most popular density map generation solutions as (Zhang et al., 2016). We smooth each ground-truth point by a fixed-size 2D Gaussian kernel.
>
> **Q12**: PDM loss for self-supervision.
>
> **A12**: Since the density values represented by the two branches are Interleaved, it does not make sense to supervise them to have similar outputs. In contrast, the expectations of the two branches should be consistent which our ECR is based on.
>
> **Q13**: Early mention of the number of layers in the transformer.
>
> **A13**: Thanks. We will explain this detail in the previous section.

---

> ### Comment · Program_Chairs · 2022-11-28
> **msg from SPC**
>
> Authors - please comment to the ethical concern raised by the reviewer.
>
> AC - if the issue remains, please bring it back to us.

---

> > ### Author Response · Authors · 2022-11-29
> > **Ethical concerns**
> >
> > Thanks for raising the concern.
> >
> > In general, this study has little ethics concerns.
> >
> > Firstly, crowd counting focuses on the overall crowd status (the total count and the rough locations of the crowd) rather than the individual status.  It does not estimate the person identity and is not linked to any identification purpose. Compared to the face detection or person-re-identification, it has much lower ethnic concern.
> >
> > Secondly, as also mentioned by the reviewer, we only use existing datasets and follow existing experimental settings for (semi-supervised) crowd counting. Thus this study doesn't introduce any new ethics issues beyond existing works of crowd counting.
> >
> > Nevertheless, thanks for the kind reminder. We will blur the faces of crowd which have a resolution higher than $24 \times 24$ to further enhance people privacy when visualizing our results in Fig. 7 of the appendix.

---

### Author Response · Authors · 2022-11-18
**Summary of Our Responses**

We thank the area chair and all reviewers for reviewing our paper.

We acknowledge the positive comments, including ‘sufficiently novel’ (Reviewers yGdj) and ‘novel’ (Reviewer ccra), ‘neat to use the uncertainty of …‘ (Reviewer yGdj), ‘most of the works … innovations’ (Reviewer FHRP) for the idea; ‘a significant improvement’ and ‘a large improvement in the semi-supervised setting’ (Reviewers yGdj), ‘good performance’ (Reviewer BsbD), ‘good results’ (Reviewer ccra), ‘performance is dramatically improved’ (Reviewer FHRP), for the performance; ‘quite clear’ (Reviewers yGdj), ‘clear and of high quality’ (Reviewer ccra), the ‘Visualizations look great’  (Reviewers yGdj), ‘a high quality manuscript’ (Reviewer ccra) for the overall presentation.

It is worth noting that the comments on our paper are polarized, with three positive (Reviewers yGdj, ccra, and FHRP) and one ‘reject’ (Reviewer BsbD). We have noticed that there are negative opinions (and also the only negative ones) from Reviewer BsbD about the correctness of using a closed form to calculate the WD distance. However, our calculation of the one-dimensional WD distance using a closed form has clear theoretical support. In the response, we have provided detailed evidence to support our calculation. We also notice that the paper recommended by Reviewer BsbD is completely different from our method, neither from the motivation nor the technology points of view. We also provide detailed responses to the comment.

Below, we respond to all reviewers' questions and concerns one by one.

---

### Decision · Program_Chairs · 2023-01-20

**Decision:**

Reject

**Justification For Why Not Higher Score:**

While the results look good and framework is reasonable, the major issue is the formulation of the loss in (4) and its underlying principle is not 1-D Wasserstein as claimed.  Further investigation is needed to explain the loss from a theoretical aspect.

**Justification For Why Not Lower Score:**

n/a

**Metareview: Summary, Strengths And Weaknesses:**

**Summary:** This paper considers the problem of predicting density maps for counting given semi-supervised datasets. It proposes a novel transformer-based architecture, a distribution-matching loss function for quantized density prediction, the use of two overlapping discretisations and a consistency loss for the two discretisations that is only applied for confident predictions in unsupervised images. The method achieves strong results in both fully and partially supervised settings, achieving state-of-the-art results. Some aspects of the design are demonstrated to be important through ablative experiments.

**Strengths:** transforming the semi-supervised problem from regression to density classification is reasonable for removing the noise. The masking strategy using uncertainty is interesting. Good performance on benchmarks.

**Weaknesses:** The following weaknesses were identified:

1. alternative explanation that the one-hot parametrization improves learnability. [yGdj]
2. can the loss in (4) be simplified since one-hot encodings are used? [yGdj]
3. needs justification for using transformer architecture. [yGdj]
4. can the first self-attention in the decoder be removed? [yGdj]
5. missing ablation studies (ECR loss w/o mask, effect of w=0.5) [yGdj]
6. should include fully supervised results. [yGdj]
7. why are BL and DM loss worse than CE? [yGdj]
8. the 1D closed from solution for Wasserstein distance used in the paper (with CDF instead of inverse CDF) requires p=1 (L1 norm). However, the paper also uses p=2, which achieves better results. [yGdj]
9. Previous work on using pixel-wise uncertainty for crowd counting. [BsbD]
10. the intervals used depends on the Gaussian kernel, which makes it less reliable for other settings. [BsbD]
11. Eq 4 is not the Wasserstein distance. [BsbD]
12. How to use the unlabeled images? [BsbD]
13. missing ablation study on the initialization method [ccra]
14. results using 40% labeled data, but Meng et al used 50% labeled data. [ccra]
15. any new semi-supervised counting in 2022? [ccra]
16. How does the interleaving dual branch structure work in semi-supervised learning? [FHRP]
17. What's the difference between the two branches? more clarifications of Fig 1 [FHRP]
18. How are the density levels generated from the annotations? [FHRP]

The authors wrote a response to address these issues.

**Summary Of Ac-Reviewer Meeting:**

During the discussion, Reviewer yGdj was satisfied with the response and also recommended some further clarifications.  The major issue for discussion was about the relationship between Eq 4 and the closed-form 1-D Wasserstein distance. As pointed out by Reviewer yGdj, substitution of the inverse CDF with the CDF in the 1-D closed-form Wasserstein is only valid for p=1 (L1 norm) case, and thus the p=2 case in (4) is not a 1-D Wasserstein. The authors offered to add a sentence in the paper: "the equation strictly holds under L1 norm. The one with L2 norm is a satisfactory approximation with better results in the experiment".

The AC noted that this replacement of inverse CDF with CDF is probably necessary to make the computation easier because the probability distributions are step-wise functions (due to the one-hot nature), and thus matching inverse CDF values is challenging.  However just performing this swap for the p=2 case, as the authors did, has no theoretical justification, although it performs better than L1.  What is the approximation principle? what exactly is being approximated? what is potentially lost in the approximation? is it an upper-bound? lower-bound?  More details should be provided to add some theoretical insight and/or justify the approximation. If (4) is not the 1-D Wasserstein, then perhaps it could be explained in other ways, e.g., as a loss on the quantiles of the distribution (suggested by the AC), or using the one-hot nature of y to derive an equivalent loss (as suggested by Reviewer yGdj).  Reviewer BsbD also expressed doubts regarding the 1-D Wasserstein and one-hot formulation.

In the end, the AC thought that the issues surrounding the loss in (4) need to be further investigated and its justification further refined. As a result the paper is not ready for publication in ICLR.